# Stunting as a Risk Factor for Asthma: The Role of Vitamin D, Leptin, IL-4, and CD23+

**DOI:** 10.3390/medicina58091236

**Published:** 2022-09-07

**Authors:** Gartika Sapartini, Gary W.K. Wong, Agnes Rengga Indrati, Cissy B. Kartasasmita, Budi Setiabudiawan

**Affiliations:** 1Doctoral Study Program, Department of Child Health, Faculty of Medicine, Universitas Padjadjaran, Bandung 40161, Indonesia; 2Department of Paediatrics, Faculty of Medicine, Chinese University of Hong Kong, Hong Kong, China; 3Department of Clinical Pathology, Faculty of Medicine, Universitas Padjadjaran, Dr. Hasan Sadikin Central Hospital, Bandung 40161, Indonesia; 4Department of Child Health, Faculty of Medicine, Universitas Padjadjaran, Dr. Hasan Sadikin Central Hospital, Bandung 40161, Indonesia

**Keywords:** stunting, asthma, vitamin D, interleukin-4, CD23+

## Abstract

Stunting, which results from chronic malnutrition, is common in children from low- and middle-income countries. Several studies have reported an association between obesity and asthma. However, only a handful of studies have identified stunting as a significant risk factor for wheezing, a symptom of asthma, although the underlying mechanism remains unclear. This article aimed to review possible mechanisms underlying asthma in stunted children. Overall, changes in diet or nutritional status and deficiencies in certain nutrients, such as vitamin D, can increase the risk of developing asthma. Vitamin D deficiency can cause linear growth disorders such as stunting in children, with lower levels of 25(OH)D found in underweight and stunted children. Stunted children show a decreased lean body mass, which affects lung growth and function. Low leptin levels during undernutrition cause a Th1–Th2 imbalance toward Th2, resulting in increased interleukin (IL)-4 cytokine production and total immunoglobulin E (IgE). Studies in stunted underweight children have also found an increase in the proportion of the total number of B cells with low-affinity IgE receptors (CD23+). CD23+ plays an important role in allergen presentation that is facilitated by IgE to T cells and strongly activates allergen-specific T cells and the secretion of Th2-driving cytokines. Stunted children present with low vitamin D and leptin levels, impaired lung growth, decreased lung function, and increased IL-4 and CD23+ levels. All of these factors may be considered consequential in asthma in stunted children.

## 1. Introduction

Stunting remains a global problem, with high incidence rates observed worldwide, particularly among low- and middle-income countries. The United Nations International Children’s Emergency Fund (UNICEF), World Health Organization (WHO), and the World Bank Group in 2021 declared that the prevalence of stunted children worldwide is 22% [1]. According to the World Bank, prevalence of stunting in lower-middle-income countries is 29.1%. The highest prevalence of stunting is on the African continent at 31.7%, followed by South-East Asia at 30.1% and the Eastern Mediterranean at 26.2% based on WHO [1].

Stunting results from chronic malnutrition. Malnourished children have lower body fat, which causes impaired lung growth and lower lung function. Moreover, malnutrition has been correlated with a higher incidence of asthma symptoms [2]. Changes in diet or nutritional status and deficiencies in certain nutrients, such as vitamin D, may affect the risk for developing asthma [2,3]. Notably, recent animal studies have reported that vitamin D deficiency in utero and in early life leads to elevated Th2 and reduced interleukin (IL)-10-secreting Treg cells, thereby potentially increasing the risk of asthma, which has been considered a Th2-mediated disease [4]. Another study found an association between vitamin D deficiency and stunting, which signifies that stunted children have lower 25(OH)D levels than controls [5]. 

Malnutrition also causes a decrease in adipocyte mass, resulting in a decrease in circulating leptin [6]. Leptin deficiency decreases the secretion of Th1 cytokines and increases the production of Th2 cytokines, including IL-4. One study showed an imbalance in the type 1/type 2 immune response among malnourished children [7]. A significant reduction in serum leptin levels resulted in a shift toward Th2 cytokine [8].

Interleukin-4, a cytokine produced by Th2 cells, is an important mediator for IgE synthesis [9]. IgE binds to FcεRI and CD23+/FcεRII and plays a role in allergic diseases. In stunted underweight children, there were abnormalities in the immune response, including increased IL-4 cytokine production, impaired T-cell responses, and an increased amount of B cells containing CD23+, which can lead to an increase in specific IgE, which then causes wheezing and asthma symptoms [10].

Hawlader et al. identified that stunting is one of the significant risk factors that predisposes rural Bangladeshi children to wheezing [2]. However, this study failed to clarify the exact mechanism by which nutritional factors affect the development of asthma, especially in stunted children. Therefore, the current article reviews the available literature to elucidate the possible underlying mechanisms causing stunting to be a risk factor for asthma among children.

## 2. Stunting and Vitamin D

Linear growth disorder is defined as a failure to achieve a person’s linear growth potential, resulting in short stature [11]. Short stature can be caused by pathological or nonpathological conditions. Stunting is the largest part of short stature, which occurs mostly due to inadequate nutrition (malnutrition), chronic infection (non-endocrine), and inadequate psychosocial stimulation [11]. A child is considered stunted if their length/height for age is less than −2 standard deviations from the median WHO Child Growth Standards curve for the same age and sex [12].

During the critical period (from the child’s conception to 2 years of age), vitamin D deficiency can increase the risk of growth retardation [13]. Vitamin D levels affect and have a significant relationship with linear growth and are important for normal growth in children [14,15,16]. On the other hand, vitamin D deficiency is correlated with decreased linear growth and stunting [17]. One study showed that stunted children aged 2–5 years in South Africa consumed less vitamin D, calcium, riboflavin, and fat than normal children [18]. A study by Bueno et al. in Brazil reported a correlation between vitamin D deficiency and short stature [5]. Moreover, lower levels of 25(OH)D have been found in underweight and stunted children aged 6–36 months in Ecuador [13]. The research on oral supplementation with vitamin D reported that the mean length/height for age z-scores in vitamin D groups were slightly higher than in the the placebo group [19].

Vitamin D plays a role in fetal lung development and maturation and maintaining lung structure and function. Prenatal vitamin D deficiency may affect fetal lung and immune system development and may be exacerbated by postnatal vitamin D deficiency [20]. Vitamin D insufficiency was also correlated with lower forced expiratory volume 1 (FEV1) and forced vital capacity (FVC) not only in adults but in children as well [21].

Stunting results in reduced lung growth, restrictive lung function, and low FVC values. Alveolar development and capillary growth are strongly influenced by postnatal vitamin A, E, and D supplementation, which are critical determinants of FVC. Small for gestational age at birth and stunting were associated with short stature in adults and reduced lean body mass, which were also correlated with low lung function [22].

A study by Håland et al. reported that low lung function at birth is correlated with an increased risk of asthma by 10 years of age [23]. Inadequate nutrition can reduce skeletal and respiratory muscle mass, leading to decreased lung function. Stunted boys and girls exhibited decreased lean body mass [24]. Lower body fat has been correlated with an increased incidence of asthma symptoms. Moreover, malnourished children may experience impaired lung growth, which also causes an increase in asthma symptoms [25].

## 3. Metabolism of Vitamin D

Vitamin D synthesis occurs mainly through sun exposure, which produces provitamin D3 that is hydroxylated in the liver and kidneys, becoming 1,25-dihydroxyvitamin D (1,25(OH)_2_D_3_), which is the active form of vitamin D that acts on target organs such as bone, immune cells, and liver cells. Vitamin D from food intake consists of two forms, namely vitamin D2 (ergocalciferol), which is sourced from plants, and D3 (cholecalciferol) which is derived from animal sources. The first step in vitamin D synthesis is the formation of vitamin D3 in the skin through the action of ultraviolet irradiation. Upon exposure to UVB light, the pathway to vitamin D in the skin begins with the breakdown of ring B 7–dehydroxycholesterol (7-DHC) to form pre-vitamin D3, which is isomerized to vitamin D3 (cholecalciferol). Cholecalciferol is bound to the vitamin D binding protein (VDBP) and transported to the liver. Cholecalciferol is hydroxylated by hepatic mitochondria and microsomal 25-hydroxylases (25-OHase), which is encoded by the CYP27A1 gene (Cytochrome P450 Family 27 Subfamily A Member 1) to calcidiol or 25(OH)D3. Calcidiol is an indicator of vitamin D status as measured by serum. Calcidiol is carried by VDBP to the kidneys, where it is hydroxylated by mitochondrial 1α-hydroxylase (1α-OHase; encoded by the CYP27B1 gene (Cytochrome P450 Family 27 Subfamily B Member 1)), resulting in the hormonally active secosteroid 1,25(OH)_2_D_3_ (calcitriol), an active form of vitamin D [26]. In addition, the concentration of 1,25(OH)_2_D_3_ is feedback-regulated: an increase in 24,25(OH)_2_D_3_ induces the synthesis of 1,25(OH)_2_D_3_, whereas Ca2+, phosphate, and 1,25(OH)_2_D_3_ alone inhibit the synthesis of 1,25(OH)_2_D_3_ [26].

## 4. Vitamin D Deficiency and Asthma

Vitamin D has been correlated with asthma given its immunomodulatory effects that involve inhibition of Th1 cell activation, modulation of Th2 cells, and increased regulatory T-cell (Treg) activity [27]. The immunomodulating potential of vitamin D has a role in the pathogenesis of asthma [21]. Bener et al. reported that vitamin D deficiency was strongly correlated with asthma, allergic rhinitis, and wheezing [28]. Esfandiar et al. reported that the risk of asthma among children suffering from vitamin D deficiency was 6.3 times higher than that in healthy children [29].

Vitamin D deficiency has been suggested to increase the incidence of asthma and allergic symptoms. Lower levels of 25(OH)D have been related to an increase in asthma prevalence and hospitalization and emergency room visitation due to asthma [30]. Al-Zayadneh et al. reported that vitamin D deficiency was correlated with asthma severity in children. This was shown by children with lower levels of vitamin D having higher GINA scores, more frequent hospital admissions, and use of systemic steroids to treat asthma exacerbations [31]. Meanwhile, Aziz et al. reported that vitamin D deficiency was associated with the length of hospitalization, an increased number of annual asthma exacerbations, and treatment in the High Care Unit [32]. The relationship between vitamin D deficiency and asthma has also been associated with decreased lung function. On average, children with vitamin D deficiency have a slightly lower FEV 1 than those with adequate vitamin D levels [33].

## 5. Possible Mechanism of Asthma in Stunted Children: The Role of Vitamin D, Leptin, IL-4, and CD23+

Until now, from the existing literature and evidence, there is no complete explanation regarding the role of stunting and the direct relationship to asthma. Hawlader et al. reported that the incidence of wheezing among rural Bangladeshi children was high, and that stunting has a significant association with wheezing. However, the mechanism that explains stunting as a risk factor for asthma is still unclear. Several studies that have been conducted have only explained several factors found in stunted children including low vitamin D levels (vitamin D deficiency), low leptin levels, and increased IL-4 and CD23+ levels, but they have not explained the role of these factors on the occurrence of asthma in stunted children.

### 5.1. Vitamin D

Among the numerous functions of vitamin D as a hormone, the classic functions are mineral balance and bone maintenance while it also act as a micronutrient and immunomodulator. As an immunomodulator, 1,25(OH)_2_D_3_ suppresses Th1 cell activation, modulates Th2 cells, and increases Treg cell activity [27]. Although asthma is a known Th2-mediated disease, the impact of vitamin D on Th2 cells reportedly differs. The impact of 1,25(OH)_2_D_3_ on Th2 cells remains controversial, with evidence suggesting that 1,25(OH)_2_D_3_ can inhibit IL-4 transcription but can also upregulate IL-4 in mouse T cells [34,35].

Vitamin D alters the balance of Th1–Th2 cytokine toward Th2, resulting in reduced secretion of Th1 cytokines IL-2 and IFN-γ and an increase in the Th2 cytokine IL-4. In contrast, in CD4+ as well as CD8 human cord blood cells, vitamin D inhibits IL-12-generated IFN-γ production and suppresses IL-4 and IL-4-induced expression of IL-13 [20]. Staeva-Vieira reported that during in vitro polarization, 1,25(OH)_2_D_3_ inhibited Th1 (IFN-γ) and Th2 (IL-4) cytokines on naïve CD62 ligand^+^CD4^+^ T cells [34]. The contradictory effect of vitamin D on Th1–Th2 dominance may be due to the effect of vitamin D on T-regulatory (Treg) cells. Vitamin D has been shown to increase Treg cell induction [20].

Vitamin D3 regulates various stages of allergic responses and promotes dendritic cells (DCs) to produce IL-10 to support Treg development. Moreover, vitamin D3 acts synchronously with steroids to induce IL-10-peripheral Treg, which can regulate the next process of the immune response by inhibiting the proliferation and differentiation of Th2 effectors through IL-10 production. Vitamin D3 has the potential to regulate Th2 immune responses by inducing CD4+CD25+ Tregs to produce regulatory cytokines, such as IL-10 or TGF-β. The direct effect of vitamin D3 with Th2 effectors is still controversial, as shown in Figure 1 [36].

Some of the beneficial effects of vitamin D on asthma include its effect on Treg cells and IL-10, which also suppresses Th2 responses [37]. Vasiliou et al. reported that perinatal vitamin D deficiency in mice promoted Th2 and decreased IL-10-secreting Treg cells [4]. The hypothesis is that low vitamin D levels found in stunted children may cause changes in the Th1–Th2 balance toward Th2 and promote allergic diseases such as asthma. although further research is still needed.

### 5.2. Leptin

Various studies have shown that malnourished children present with low levels of growth hormone, leptin, and prolactin, all of which promote thymic growth and function [38]. Leptin is a hormone/cytokine derived from adipocytes and links the nutritional conditions with neuroendocrine and immune systems [8]. Leptin has direct and indirect effects on the number and function of T cells, increasing the number of cells and the Th1 and Th17 cytokines while also inhibiting the production of Th2 cytokines and Treg proliferation [6]. Adipocyte-derived leptin secretion has been associated with normal regulation of metabolic function and a balance between Th1 and Th2/Treg cells that causes the suppression of immune and autoimmune responses [8]. 

Malnutrition causes a decrease in adipocyte mass, which results in a decrease in circulating leptin [6]. Malnourished children develop changes in CD4+ and CD8+ T cell numbers and functions mediated by leptin [6]. Mice with leptin deficiency have increased production of Th2 cytokines, including IL-4 and decreased secretion of IL-2, IFN-γ, and TNF-α. This suggests that leptin is involved in disruption of the Th1 and Th2 cytokine regulation. A significant reduction in serum leptin levels results in a shift toward Th2 cytokine production [8]. Therefore, low leptin levels in stunted children may cause changes in the Th1/Th2 balance toward Th2 cells, thereby increasing IL-4 production.

### 5.3. Interleukin (IL)-4

IL-4 is produced by Th2 cells, mast cells, basophils, eosinophils, and the alveolar macrophages [39]. IL-4 is a pleiotropic cytokine that binds to the IL-4 receptor expressed by T lymphocytes, B lymphocytes, mononuclear phagocytes, eosinophils, pulmonary fibroblasts, endothelial cells, bronchial epithelial cells, and smooth muscle cells. IL-4 is involved in the different mechanisms leading to allergic airway disease. Transcription factors are activated by the IL-4/IL-13 signaling cascade (Figure 2) [39].

Malnourished children have severely impaired IL-2 and IFN-γ production; however, they demonstrate increased IL-4 production by CD4+ and CD8+ cells. The study of Martinez et al. showed an imbalance in the type 1/type 2 immune responses among malnourished children (i.e., the cytokine pattern is skewed toward a Th2 response) [7]. The increase in IL-4 levels among malnourished children could be a risk factor for developing asthma although further research is still needed.

### 5.4. CD23+

A low-affinity receptor for IgE, namely FcεRII or CD23+, presents an important role in regulating IgE responses. The regions of CD23+ responsible for its interaction with several ligands, which include IgE, CD21, major histocompatibility complex class II, and integrins, were discovered [40]. 

The interaction between IgE and the CD23+ remains unclear and has not been comprehended completely compared to the interaction between IgE and the high-affinity IgE receptor (FcεRI), even though CD23+ plays many significant roles in the allergic inflammatory process. One of the important roles of CD23+ is presenting allergen to T cells, which are facilitated by IgE. IgE-facilitated allergen presentation potentially activates allergen-specific T cells and secretes Th2-driving cytokine [41]. 

Overall, the expression of CD23+ surface density on B cells among allergic patients will increase in the presence of high levels of allergen-specific IgE. This leads to enhance IgE-facilitated allergen presentation and allergen-specific T-cell activation [41]. T-cell-mediated allergic inflammation induced by IgE-facilitated allergen presentation might be controlled by CD23+ surface density on B cells and allergen-specific T-cell activation [42]. The role of CD23+ in allergic airway disease is explained in Figure 3 [43]. 

Studies on stunted underweight children also found an increase in the proportion of the total number of B cells with CD23+, impaired T-cell responses, and increased levels of total IgE and IL-4 [10]. Theoretically, the important roles of CD23+ is IgE-facilitated allergen presentation to T cells, strong activating allergen-specific T cells, and secreting Th2-driving cytokines. Therefore, further research is needed to determine whether increased CD23+ levels increase the risk of developing allergies in stunted children.

Therefore, the possible mechanisms of asthma in stunted children described above can be summarized as in Figure 4.

## 6. Conclusions

In conclusion, this review showed that stunted children exhibit low vitamin D and leptin levels, decreased lean body mass, impaired lung growth, decreased lung function, and increased IL-4 and CD23+ levels. All these factors may be considered to play a prominent role in the occurrence of asthma in stunted children. However, further research is required on this topic.

## Figures and Tables

**Figure 1 medicina-58-01236-f001:**
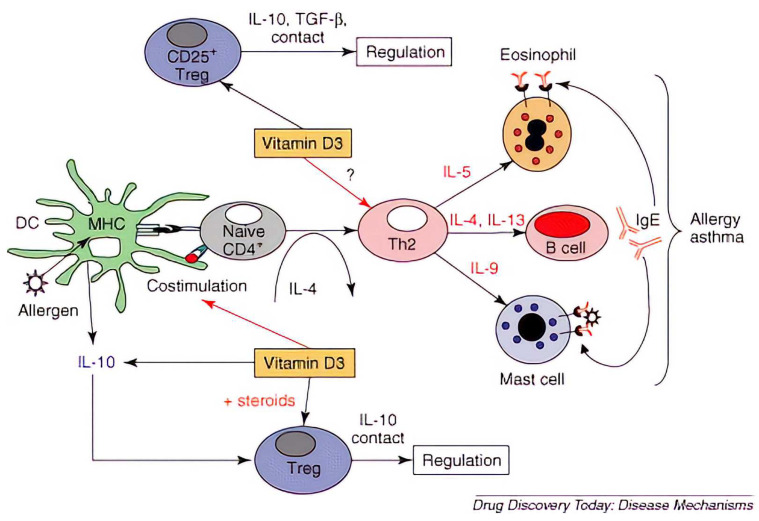
Effects of vitamin D3 on Th2-mediated allergic diseases. Reprinted with Permission from Ref. [36]. Copyright 2022 Drug Discovery Today: Disease Mechanisms.

**Figure 2 medicina-58-01236-f002:**
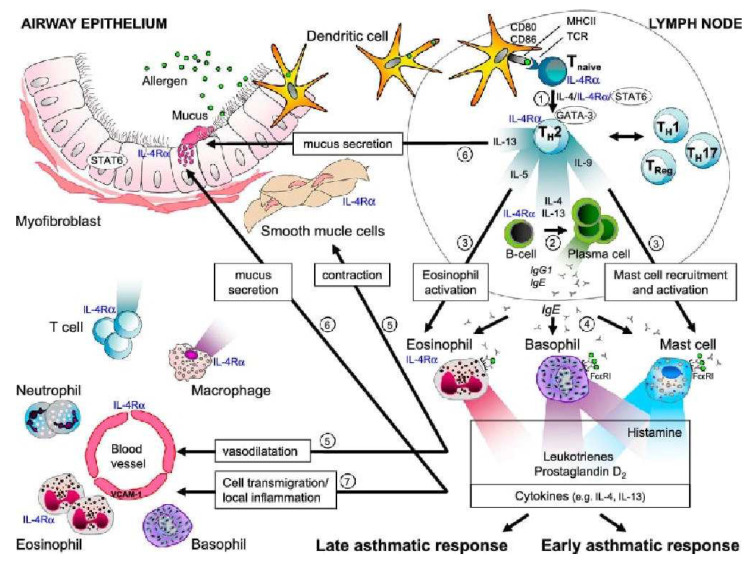
Role of IL-4 in asthma and allergy. Arrow 1 shows the role of IL-4 in the differentiation of T cells into Th2 cells. Arrow 2 shows the class-switching isotype of B cells that synthesize IgE, which is mediated by IL-4 and IL-13. Arrow 3 shows the effect of IL-5 on eosinophils and that of IL-9 on mast cell function. Arrow 4 indicates that IgE produced by plasma cells binds to the IgE receptors on mast cells, basophils, and eosinophils. Upon allergen exposure, IgE crosslinking causes an immediate release of histamine, leukotrienes, and prostaglandins (after a few minutes) and the production of other cytokines (after a few hours). Arrow 5 shows that histamine, leukotrienes, and prostaglandins induce smooth muscle contractions and vasodilation. Arrow 6 indicates that cytokine IL-13, together with IL-4, causes mucus secretion and goblet cell hyperplasia. Arrow 7 shows that IL-4 and IL-13 contribute to the recruitment of inflammatory cells by upregulating the expression of VCAM-1, which facilitates the transmigration of eosinophils, T cells, monocytes, and basophils in response to chemokines produced by mast cells (the late asthma response) [39]. AHR, airway hyperresponsiveness; FcɛRI, Fc epsilon RI, high-affinity receptor for the Fc region of immunoglobulin E (IgE); IL-4Ra, interleukin-4 receptor a; Stat6, signal transducer and activator of transcription 6. Reprinted with permission from Ref. [39]. Copyright 2022 American Thoracic Society.IL-4 is the main cytokine in the sequential development of allergic inflammatory reactions. It mediates an important proinflammatory function in asthma by inducing IgE isotype switch and IgE secretion via B lymphocytes and vascular cell adhesion molecule-1 (VCAM-1) expression, increasing eosinophil transmigration through the endothelium, mucus production, and Th2 lymphocyte differentiation and promoting cytokine release. IL-4 increases IgE-mediated immune responses by regulating IgE receptors, namely the low-affinity IgE receptor (FcεRII or CD23+) on B lymphocytes, mononuclear phagocytic cells, and the high-affinity IgE receptor (FcεRI) on mast cells and basophils. IL-4 contributes to airway obstruction in asthma by inducing mucin gene expression and increasing mucus secretion and other inflammatory cytokines from fibroblasts, which contribute to the inflammatory process and pulmonary remodeling [39].

**Figure 3 medicina-58-01236-f003:**
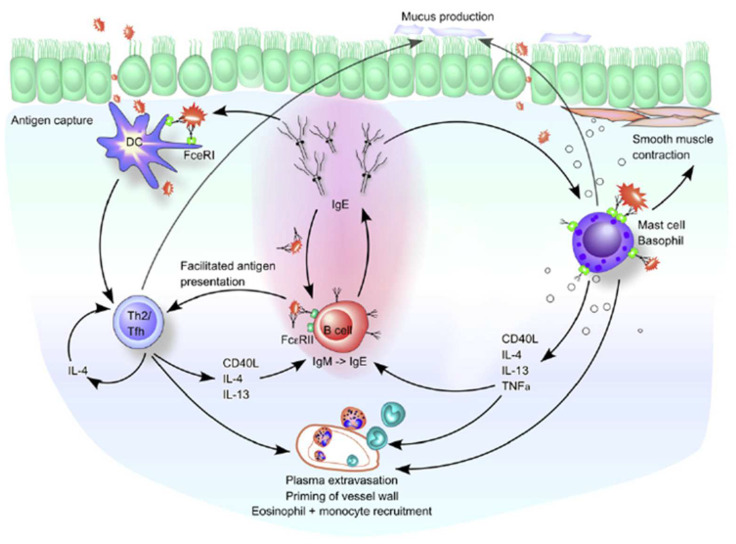
Role of CD23+/FcεRII and IgE in allergic airway diseases. DCs initially take up inhaled antigens in an atopic environment and result in a predominant induction of Th2 cells. Th2 cells potentially induce IgE-producing B cells through the production of IL-4 and IL-13. The resulting IgE then binds to its high-affinity (FcεRI) and low-affinity (CD23+/FcεRII) receptors on the airway mucosa cells. The binding of FcεRI-bound IgE to the allergen on mast cells and basophils causes degranulation and release of inflammatory mediators and cytokines, leading to immediate hypersensitivity symptoms. The late-phase response begins when the released Th2 cytokines (IL-4, IL-5, and IL-13) recruit and activate the inflammatory cells, such as monocytes and eosinophils. IL-4 and IL-13 also cause excessive mucus production by goblet cells. Elevated local IgE levels induce FcεRI expression on DCs and stabilize CD23+/FcεRII on B cells. Therefore, in a Th2-skewing environment, these cell types can take up allergen-IgE complexes and present the allergen to T cells. For B cells bearing CD23+/FcεRII, this is called facilitated antigen presentation because it occurs in a noncognate manner. Reprinted with permission from Ref. [43]. Copyright 2022 Journal of Allergy and Clinical Immunology.Chary et al. reported that children with asthma had a higher proportion of B cells with CD23+ expression than controls [44]. CD23+ expression of lymphocytes among children with asthma (due to allergic response) increases and is positively associated with serum IgE levels [40]. Excessive expression of CD23+ in patients with extrinsic asthma results from excessive IL-4 production by allergen-specific T cells [45]. The expression of CD23+ has been associated with allergic diseases. Anti-CD23 therapy may be a promising candidate therapy for allergic diseases in the future. Anti-CD23 might induce tolerance, block antigen presentation, suppress T-cell differentiation, and diminish APC activation in allergic reactions [46].

**Figure 4 medicina-58-01236-f004:**
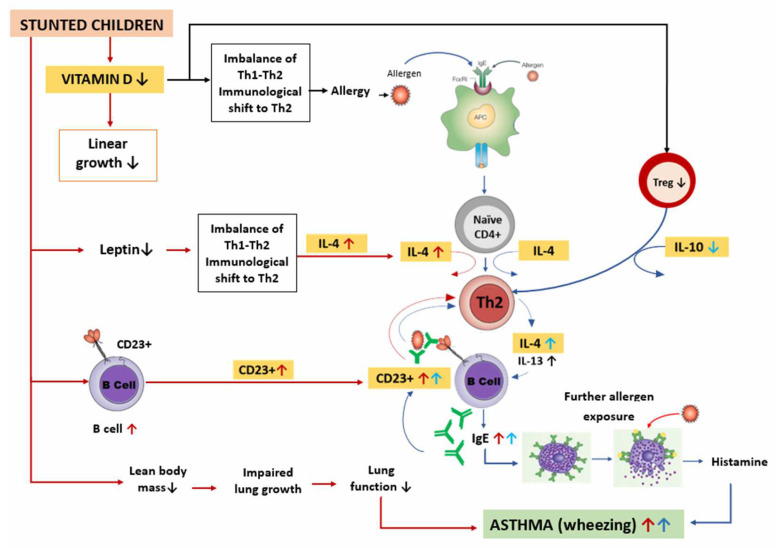
Possible mechanism of asthma in stunted children. Stunted children have lower vitamin D levels, resulting in decreased linear growth. Low vitamin D levels cause changes in the Th1–Th2 balance toward Th2, promoting allergic diseases including asthma. The pathophysiology of asthma begins with exposure to allergens captured by APC, which are then presented to naïve CD4+ cells. With the help of IL-4, naïve CD4+ cells differentiate into Th2 cells that produce IL-4 and IL-13. Thereafter, B lymphocytes differentiate into plasma cells that produce IgE and attach to the mast cells. Further exposure to allergens and crosslinking of allergens with IgE occur on the outer surface of membrane of mast cells, resulting in degranulation, histamine release, and bronchoconstriction, leading to asthma symptoms such as wheezing. In addition, when vitamin D levels are low, the number of Tregs decreases, thereby lowering IL-10 production. This causes an increase in Th2 activity and IL-4, resulting in an increase in IgE, which further stabilizes the CD23+ B cells. B cells take up allergen–IgE complexes and present the allergen to Th2 cells, thereby increasing the Th2 response. In stunted children, there is a decrease in lean body mass that causes impaired lung growth and decreased lung function. In addition, low leptin levels cause changes in the Th1/Th2 balance toward Th2 cells, thereby increasing IL-4 production. B cells also increase, causing an increase in CD23+ levels. All these factors contribute to asthma occurrence in stunted children.

## Data Availability

Data sharing not applicable.

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
