# Peer review of "Stunting as a Risk Factor for Asthma: The Role of Vitamin D, Leptin, IL-4, and CD23+"

_medicina, 2022, doi:10.3390/medicina58091236_

Round 1

Reviewer 1 Report

Summary 

Stunting as a risk factor for asthma: Potential mechanisms 2 Gartika Sapartini 1,* , Gary WK Wong 2 , Agnes Rengga Indrati 3 , Cissy B. Kartasasmita 4 and Budi Setiabudiawan 5

This review article gives a summary of the relationship between stunting and asthma It shows how the innate immune system is dysregulated as a result of chronic malnutrition. The article shows the role of vitamin D deficiency from stunting and the direct relationship to asthma 

Strength 

A well-written review article with diagrams illustrating the pathophysiology and immunology of vitamin D deficiency from stunting and the role in the production of pro inflammatory markers which is associated with asthma 

A robust reference base 

Excellent diagrams with explanation 

Weakness 

A review article at the most. 

The title does not do justice to the content 

I think a better title might perhaps be 

The role of Vitamin D deficiency in the development of asthma

The role of stunting and the direct relationship to asthma was not well elucidated by the authors 

The manuscript is very long and verbose 

Reviewer 2 Report

This review article is focused on identifying possible mechanisms that could explain how stunting is a risk factor for wheezing and asthma.  The authors propose reduced Vitamin D and leptin, as well as increased expression of CD23+ B cells are the most likely mechanisms for stunting leading to asthma.  

While Vitamin D, leptin and B cells seem to be the 3 mechanisms discussed in the review article the document did not clearly state these 3 mechanisms, and the headings throughout the document did not highlight these 3 areas.  The authors should revise this document to emphasize the potential mechanisms more clearly. 

Throughout the document the form of Vitamin D varies. Please include a section describing the synthesis pathway of Vitamin D to orient the reader.  Does stunting alter Vitamin D synthesis?

General comment - the authors point to high prevalence of stunting in Africa, Asia and Oceania.  What is the prevalence of asthma in these regions? Does it correlate?  A quick internet search points to Africa having a growing number of asthmatics that is likely underreported but this does not seem to be the case for Asia. Why would a similar incidence of stunting not associate with a same prevalence of asthma? Is there a mechanism at play, such as differing allergen exposure? fungal, hdm, etc.?

Introduction section - The authors have overstated the literature in a few places. It is recommended they rephrase the following statements. There is no mention of B cells and CD23 in the introduction, this should be included.

Line 44 - Vitamin D has been associated with asthma but has not been shown to increase the risk, revise

Line 46 - clarify that Vitamin D deficiency results in elevated Th2

Line 59-61 - clarify the purpose of this review. Is it to focus on evidence that stunting is risk factor for asthma? or to identify possible mechanisms for why stunting is a risk factor for asthma? or both? 

Linear growth disorder in stunting - does this heading title make sense?  This section needs to either focus on defining linear growth disorder or vitamin D, but not both.

The authors should include and discuss literature showing that oral supplementation with Vitamin D does not improve linear growth.  PMID:33305842

Stunting and lung function

This section would benefit from being included with the previous section - Linear growth disorder in stunting.

Please describe how impaired lung function increases risk of asthma.  The reference included to state that impaired lung growth causes an increase in asthma symptoms does not show that. The authors conclude that asthma is associated with lower body fat and not impaired lung function.

Immune system in stunting

This section should include more information about the effects of the immune system in stunting.  Specifically in relation to the information provided in Figure 1 that is not discussed such as effects on epithelial cells, granulocytes, mononuclear phagocytes, NK cells, ILCs. Alternatively, consider removing the figure or supplementing with a simpler diagram.

Line 112 - this appear to be a new section focused on leptin, but it does not have a heading. Includes discussion of leptin and stunting. Include more information about leptin and its link to asthma.

Line 130-135 - repetitive and seems out of place

Vitamin D deficiency and asthma

Line 137-139 - include a reference

Line 142-143 - are you sure increasing Tregs following Vitamin D treatment also increases a shift to Th2? Later on line 148-149 and Figure 2 you state that increased Tregs reduces Th2. Please resolve this.

Line 160-161 - Please clarify if lower vitamin d is related to increased asthma prevalence or poor management of existing asthma (check reference)

Pathomechanism of asthma: the role of IL-4 and CD23+

This section needs a better transition and a better title - maybe reworking this section to better focus on CD23 and asthma and stunting.  There is a lot of discussion about all of the ways IL-4 influences asthma, yet the authors focus on CD23 as the mechanism that connects stunting and asthma.  Why not IgE, mast cell numbers, mucus secretion, VCAM-1 or eosinophil levels? 

Line 172-173 : Clarify what this sentence is saying.

anti-CD23 has been considered as a target for treating asthma by acting to reduce IgE and prevent symptoms, please discuss this. 

Figure 4 - this figure highlights the effects of FcER1, particularly 4b which is not something focused on by the authors.  Please modify figure to reflect the emphasis on CD23.

Line 243-244 - what is the reference showing CD23 levels in stunted children?

Possible mechanism of asthma in stunted children

This section is very repetitive and would be better if broken up into each possible mechanism being discussed separately.

Conclusions

line 342 - rephrase this study, to this review

Reviewer 3 Report

The manuscript by Sapartini, G. et al. “Stunting as a risk factor for asthma: Potential mechanisms” is a review of literature that summarized the possible mechanism in stunting as a risk factor for asthma. Stunting, which results from chronic malnutrition, is common in children from low- and  middle-income countries. Stunting has also identified as a risk factor for wheezing, a symptom of asthma. This study aimed to review potential mechanisms underlying asthma in stunted children. The effect of overall, changes in diet or nutritional status and deficiencies in certain nutrients, such as vitamin D, can increase the risk of developing asthma and TH2 immune component was also discussed.

1.     Introduction paragraph 1: The data presented here in this paragraph is from old report. A newer data is available with the 2021 report. It will be good to revise these figure with the most up-to-date data publicly available and also update the references [1].

2.     References cited are old and not recent, only one reference is 2021. There are a several important paper published that discussed the updated research, however they were neither cited in this study nor they were discussed. It would be better to cite the recent literature to make this review up-to-date summary of the current literature.

3.     It would be good to discuss the dietary intervention/ nutritional supplement that can be employed to mitigate the stunting in early childhood life.

Round 2

Reviewer 1 Report

Thank you .

Author Response

Thank you for your suggestion. The authors also agree with the title of this manuscript "Stunting as a risk factor for asthma: the role of vitamin D, leptin, IL-4, and CD23+"

Thank you for your comment dan response. 

Reviewer 2 Report

The authors have responded to most of the concerns brought forward improving the readability of the document. There are still some revisions that should be made.

Lines 40-41. Please clarify 1) Why did the percent prevalence change between the 1st and 2nd submission for the countries listed? 2) Why is the prevalence of asthma 29.1% across developing countries, but the 3 countries listed have much higher incidence?

Line 47. The authors have not made this change. The statement still needs to be changed to state ...early life leads to elevated Th2 and reduced..

Lines 95-98. The same statement describing postnatal vitamins impact alveolar development and capillary growth is provided twice. Please remove one.

Lines 99-101. Please rephrase this sentence. It is unclear.

Line 151. Please use the heading provided in the author's reply of "Possible mechanism of asthma in stunted children" in place of the "Possible mechanism of asthma in stunting..." 

Line 152. A paragraph outlining the possible mechanisms would be helpful.

Reviewer 3 Report

Authors have addressed all of my comments. 

Thank you

Author Response

Thank you for your comment dan response.